# Epigenetic Regulation of β-Globin Genes and the Potential to Treat Hemoglobinopathies through Epigenome Editing

**DOI:** 10.3390/genes14030577

**Published:** 2023-02-25

**Authors:** Letizia Fontana, Zoe Alahouzou, Annarita Miccio, Panagiotis Antoniou

**Affiliations:** 1Laboratory of Chromatin and Gene Regulation during Development, INSERM UMR 1163, Imagine Institute, Université Paris Cité, F-75015 Paris, France; 2Genome Engineering, Discovery Sciences, BioPharmaceuticals R&D Unit, AstraZeneca, 431 50 Gothenburg, Sweden

**Keywords:** globin gene regulation, epigenome editing, β-hemoglobinopathies

## Abstract

Beta-like globin gene expression is developmentally regulated during life by transcription factors, chromatin looping and epigenome modifications of the β-globin locus. Epigenome modifications, such as histone methylation/demethylation and acetylation/deacetylation and DNA methylation, are associated with up- or down-regulation of gene expression. The understanding of these mechanisms and their outcome in gene expression has paved the way to the development of new therapeutic strategies for treating various diseases, such as β-hemoglobinopathies. Histone deacetylase and DNA methyl-transferase inhibitors are currently being tested in clinical trials for hemoglobinopathies patients. However, these approaches are often uncertain, non-specific and their global effect poses serious safety concerns. Epigenome editing is a recently developed and promising tool that consists of a DNA recognition domain (zinc finger, transcription activator-like effector or dead clustered regularly interspaced short palindromic repeats Cas9) fused to the catalytic domain of a chromatin-modifying enzyme. It offers a more specific targeting of disease-related genes (e.g., the ability to reactivate the fetal γ-globin genes and improve the hemoglobinopathy phenotype) and it facilitates the development of scarless gene therapy approaches. Here, we summarize the mechanisms of epigenome regulation of the β-globin locus, and we discuss the application of epigenome editing for the treatment of hemoglobinopathies.

## 1. Introduction

### 1.1. Hemoglobins

Hemoglobin (Hb), the main component of mature red blood cells (RBCs), is a tetrameric metalloprotein composed of four polypeptide globin helicoid chains (two α-like and two β-like globins), a heme molecule and a ferrous ion (Fe). Different variants of Hb, with a different composition of α- and β-like chains, are produced throughout life in a stage-specific manner [1] with two Hb switches taking place during life, a primary switch from embryonic to fetal and a second switch from fetal to adult that occurs shortly after birth. Fetal Hb (HbF) is replaced by adult Hb (HbA), which is the predominant Hb during adult life (>95% of total Hb), while HbF and HbA_2_ are normally expressed at low levels (~1%) [2,3,4] (Figure 1). Even though the different Hb types present a very similar structure with only a few amino acid differences in the globin chains, they have a different O_2_ affinity. Embryonic Hb (Hb Gower 1, ζ_2_ε_2_; Hb Gower 2, α_2_ε_2_; and Hb Portland 1, ζ_2_γ_2_) and fetal Hb (HbF, α_2_γ_2_) exhibit a higher O_2_ affinity than the adult one (HbA, α_2_β_2_; and HbA_2_, α_2_δ_2_) [5,6].

### 1.2. Globin Gene Regulation

Human globin genes are organized in two different clusters, the α- and the β-globin loci, located in chromosomes 16 and 11, respectively. The human α- and β-globin locus genes are arranged in a 5′ to 3′ direction according to their sequential expression during development (Figure 1). Human globin genes are expressed in an erythroid- and developmental stage-specific manner during life. The regulation of gene expression is accomplished at three levels: by proximal regulation through transcription factors (TFs) that bind to globin gene promoters and allow their expression [7,8,9]; by distal regulation through the locus control regions (LCR) of α- and β-globin loci that loop with globin promoters and allow developmental stage-specific expression via their DNase I hypersensitive sites (HS) that act as enhancers or insulators [4,10,11,12,13,14]; and lastly, by epigenetic modifications associated with high or low gene expression levels (see “Epigenetic regulation of the β-globin genes”; Figure 2).

### 1.3. The Fetal-to-Adult Hb Switching

The fetal-to-adult Hb switch results in the silencing of γ-globin genes and the activation of the β-globin gene. The specific binding of TFs to the β-like globin genes and the chromatin remodeling through LCR-mediated looping formation is required for the regulated expression of β-like globin genes. The chromatin looping drives the γ- or β-globin gene expression in fetal and adult erythroblasts, respectively. The looping is mediated by a protein complex and facilitates the juxtaposition of the LCR to the γ- or β-globin gene promoter, thus promoting their expression [32,33,34,35,36].

Numerous TFs and co-factors are involved in the repression or activation of the *HBG* genes (Figure 2). Genome-wide association studies have linked single-nucleotide polymorphisms (SNPs) with abnormally elevated HbF levels [37,38]. The location of these SNPs facilitated the identification of one of the major transcriptional repressors of γ-globin genes, B-cell lymphoma/leukemia 11A (BCL11A) [39] that suppresses gene expression by binding to the γ-globin promoters [40].

Different isoforms of BCL11A are expressed in adult and fetal erythroblasts, thanks to alternative mRNA splicing. Larger forms (XL and L) are expressed in adult cells, whereas shorter forms (S and XS) are expressed in fetal cells, indicating the stage-specific repressive action of larger BCL11A isoforms [41]. BCL11A knockdown experiments showed an increase in γ-globin mRNA and HbF levels in erythroid cells derived from human HSPCs, with no impairment of the erythroid differentiation. BCL11A interacts and recruits other factors/complexes that act as co-repressors, such as the nucleosome remodeling deacetylase (NuRD) complex and polycomb repressive complex 2 (PRC2 containing Enhancer of zeste homolog 2 (EZH2), Embryonic ectoderm development (EED), and SUZ12) [26,42,43]. More recently, the zinc finger and BTB domain containing 7A (ZBTB7A), also known as lymphoma/leukemia-related factor (LRF), was identified as another major γ-globin repressor that binds to the γ-globin promoters, recruits NuRD and downregulates gene expression in a similar yet independent way from BCL11A [44,45]. The NuRD polymeric protein complex, upon recruitment at the γ-globin genes by BCL11A or LRF, is thought to act through its components, such as histone deacetylases (HDACs) [43] to silence gene expression.

### 1.4. Beta-Hemoglobinopathies

Beta-hemoglobinopathies are monogenic diseases caused by mutations in the β-globin locus that affect the synthesis, structure or properties of HbA. β-thalassemia is caused by mutations in the β-globin locus that reduce (β^+^) or abolish (β^0^) the production of adult β-globin chains composing the HbA tetramer. This leads to the precipitation of uncoupled α-globin chains, ineffective erythropoiesis, erythroid cell death by apoptosis and anemia [46]. In SCD, an A>T mutation in the β-globin gene (*HBB*) causes the substitution of valine for glutamic acid at position 6 of the β-globin chain (β^S^) that is responsible for deoxygenation-induced polymerization of the sickle hemoglobin (HbS). This primary event drives ineffective erythropoiesis, apoptosis of erythroid precursors, red blood cell (RBC) sickling, hemolysis, vaso-occlusive crises and multi-organ damage, often associated with severely reduced life expectancy [47].

Allogeneic HSC transplantation is the only curative therapy for β-hemoglobinopathies. However, the absence of sibling donors and the risk of immunological complications prevent its use in a large fraction of patients. The clinical severity of β-hemoglobinopathies is alleviated by the persistent expression of fetal γ-globin chains in adulthood, e.g., in hereditary persistence of fetal hemoglobin (HPFH). In β-thalassemia, elevated fetal γ-globin levels lower the imbalance between α- and non-α globin chains, reduce the formation of α-globin precipitates and thus improve the clinical phenotype. In SCD, fetal γ-globin exerts a potent anti-sickling effect by replacing the sickle β-globin in the Hb tetramers [48]. Therefore, the γ-to-β-globin switch is the subject of extensive study, as understanding the underlying mechanisms is crucial to develop novel therapeutic strategies to combat this multifaceted disease group.

## 2. Epigenetic Regulation of the β-Globin Genes

Epigenetic modifications, which are defined as chromatin alterations that result in a determined expression pattern without changing the genetic sequence, have been shown to occur during the fetal-to-adult hemoglobin switching (Figure 3). These potentially heritable modifications, which include DNA methylation and histone modifications (acetylation and methylation), can be cell-, tissue- and developmental-stage specific. Epigenetic modifications change the chromatin structure and DNA accessibility, and consequently affect gene expression, as epigenetic modifiers co-operate with TFs to repress or activate gene expression (Figure 2).

### 2.1. DNA Methylation

The promoters of many human genes (including globin genes) contain Cytosine-phosphate Guanine (CpG) sites, which occur at a high frequency in regions known as “CpG islands”, and are amenable to epigenetic modifications. DNA methyltransferases (DNMT) deposit methyl groups in the fifth carbon atom of the cytosine of a CpG site. DNMT3A and DNMT3B mediate the de novo deposition of methyl groups in a non-methylated 5′-CpG-3′ and symmetrically to the non-methylated 3′-GpC-5′ of the complementary strand [49]. DNMT1 is responsible for the maintenance of the methylation of asymmetrically semi-methylated CpG sites upon DNA replication [50]. The ten-eleven translocation methyl-cytosine dioxygenase (TET) proteins TET1, TET2 and TET3 are responsible for DNA demethylation [51,52]. TET is responsible for the removal of the methyl group from a 5-methyl-Cytosine (5-mC) through the generation of a 5-hydroxy-methyl-cytosine (5-hmC) intermediate [51,52]. High DNA methylation levels normally characterize transcriptional inactive promoter regions, while hypomethylated promoters and promoters bearing the 5-hmC intermediate are generally highly expressed [53]. However, CpG methylation is not always associated with gene repression, such as when it occurs at enhancer regions. For this reason, some studies suggest that CpG methylation denotes active enhancer regions and might be required for depositing the active H3K27 histone acetylation mark (see below) [54].

In the γ-globin promoters, CpG sites span a 500 nt region centered on the transcription start site (−162, −53, −50, +6, +17, +50, CpG sites) [55,56,57] (Figure 3). Studies in erythroid cells obtained from human hematopoietic stem and progenitor cells (HSPCs) demonstrated that the γ-globin promoters were significantly hypomethylated in cord blood and fetal liver samples compared to adult bone marrow samples [55,56,57]. Conversely, the DNA methylation levels of the adult β-globin promoter were higher in fetal cells than in adult samples [55]. Moreover, an abnormal DNA hypomethylation pattern in the γ-globin promoters was associated with an increased HbF expression in β-thalassemia patient cells as compared to healthy donors [56] and in individuals with HPFH mutations [58]. By contrast, in a non-human primate baboon model, 5hmC levels positively correlated with γ-globin expression, although the 5hmC levels were substantially lower than 5Mc [53]. Interestingly, a missense mutation in the *DNMT1* gene found in patients with HbE/β-thalassemia, leads to lower stability and enzymatic activity loss of DNMT1. This was associated with the reactivation of HbF due to the diminished recruitment of DNMT1 to the γ-globin promoters and the reduced methylation levels in CpG sites in the γ-globin promoters (−53, −50, +6, +17, and +50) of erythroid cells differentiated from patient HSPCs [59].

### 2.2. Histone Modifications

The DNA of eukaryotes is wrapped around histone octamers, i.e., nucleosomes, forming the chromatin. A 147 bp-long DNA fragment is wrapped around a histone octamer, composed of two copies of histone (H) 2A, H2B, H3 and H4, while histone H1 binds to the linker DNA between two nucleosomes. A variety of post-translational modifications (e.g., methylation, acetylation and phosphorylation) in different amino acid residues of the histones have been identified and associated with distinct regulatory functions [60,61,62,63].

#### 2.2.1. Histone Methylations

The transfer of methyl groups to histone amino acid residues by methyl-transferase proteins (lysine methyltransferase (KMT) and protein arginine methyltransferase (PRMT)) is associated either with an open chromatin state and gene activation, or to a compact chromatin state and gene repression. For instance, the H3 lysine 4 monomethylation, dimethylation or trimethylation (H3K4me1, H3K4me2 and H3K4me3, respectively) and the asymmetric H4 arginine 3 dimethylation (H4R3me2a) mark transcriptionally active genomic regions (with H3K4me1 typically marking enhancers, H3K4me2 marking promoters and enhancers, and H3K4me3 marking promoters). On the other hand, the deposition of methyl, dimethyl or trimethyl groups on H3 lysine 9 or lysine 27 (H3K9me1, H3K9me2, H3K9me3, H3K27me2 and H3K27me3, respectively) or symmetric H4 arginine 3 (H4R3me2s) or H4 lysine 20 (H4K20me3) is associated with transcription inactivation [64]. Interestingly, the simultaneous presence of activating H3K4me3 and repressive H3K27me3 characterizes a bivalent chromatin state typical of genes poised for rapid activation [61]. These modifications can be removed by demethylases (e.g., lysine-specific histone demethylase 1 (KDM1A), also known as lysine-specific demethylase (LSD1)).

The role of histone methylation in the regulation of the β-globin genes has been studied for decades (Figure 3). The developmental-stage specific pattern of histone modifications along the β-globin locus was analyzed by comparing human embryonic cell-, fetal liver- and bone marrow-derived erythroblasts. The three sources have a distinct globin expression profile corresponding to an embryonic/fetal, fetal and adult state, respectively. Accordingly, the presence of the active H3K4me3 mark is correlated with gene activity in the different cell types [65]. Studies have revealed a correlation between the type of histone methylation and the activation or repression of globin genes [66,67,68]. For instance, in human adult erythroid cells, the activating H3K4me2/me3 and the repressive H3K9me1 marks are present at the β- and γ-globin promoters, respectively [43,66]. Interestingly, upon administration of tranylcypromine (TCP), a KDM1A inhibitor [69], the activating H3K4me2 mark increased in the γ-globin but not in the β-globin promoter along differentiation, suggesting that KDM1A is responsible for the repression of the fetal genes by removing the activating H3K4me2 mark.

Additional proteins are involved in the determination of the histone methylation status of the β-globin locus. In particular, the expression of *EHMT1/2* as well as the global H3K9me2 levels decrease along the erythroid differentiation in cells derived from adult HSPCs [70]. Pharmacological inhibition of EHMT1/2 (using UNC0638) or shRNA-mediated knockdown of *EHMT1/2* led to decreased levels of H3K9me2 and, consequently, HbF reactivation [70,71]. However, the reactivation of HbF observed upon EHMT1/2 downregulation was associated with an additional activating event, the acetylation of H3 lysine 9 (H3K9Ac) at the *HBG1/2* promoters, which is typically associated with gene activation (see paragraph “Histone acetylations”) [70]. Promisingly, UNC0638 also led to high levels of HbF expression in erythroid cells from β-thalassemia patients [72]. FTX-6058, and inhibitor of EDD, was also tested in healthy volunteers and induced high HbF levels without major side effects (NCT04586985).

Methylation of arginine 3 on H4 also plays an important role in globin gene regulation. In human erythroid cell lines, PRMT5 induces symmetric H4R3me2 in the γ-globin promoters [73]. The developmental-stage specific pattern of H4R3me2s marks along the β-globin locus was observed in primary erythroid cells. H4R3me2s levels in the γ-globin promoters were higher in human bone marrow-derived erythroid cells compared to cord blood-derived erythroid cells [73]. Conversely, H4R3me2s marks and PRMT5 were absent from the β-globin promoters of human bone marrow-derived erythroid cells. Small-molecule inhibition of PRMT5 (using adenosine-2′,3′-dialdehyde (Adox)) in erythroid cells derived from adult bone marrow HSPCs reduce H4R3me2s levels at the γ-globin promoters with a concomitant HbF reactivation. PRMT5 is thought to be recruited at the γ-globin promoters by LYAR. In fact, upon LYAR knockdown, both PRMT5 occupancy and H4R3me2 levels decrease at the γ- globin promoters [74].

An interplay between the different histone methylations and between histone methylations and DNA methylation has been reported at the β-like globin promoters. In human erythroid cell lines, PRMT5-induced symmetric H4R3me2 in the γ-globin promoters is associated with the recruitment of DNMT3A that methylates close CpG sites, highlighting the interplay between histone and DNA methylation [73]^.^ Furthermore, in human erythroid cell lines, PRMT5 also induces the deposition of additional repressive histone methylation marks in the γ-globin promoters, such as H4K20me3, H3K9me3 and H3K27me3, likely by recruiting other repressor complexes such as KMT5B (also known as SUV4-20h1), a lysine methyltransferase that deposits the repressive H4K20me3 mark. The H4K20me3 mark was higher at the embryonic/fetal ε- and γ-globin promoters in human bone marrow-derived erythroblasts as compared to cord blood-derived erythroid cells. Accordingly, shRNA-mediated downregulation of SUV4-20h1 led to γ-globin gene reactivation in adult cells [64].

#### 2.2.2. Histone Acetylations

The deposition of acetyl groups on histone lysine residues loosens the interactions between histones and DNA due to a neutralization of the positive histone charge [75,76]. As a result, chromatin is in an open state, and TFs can bind to the DNA resulting in transcription activation. The proteins that are responsible for histone acetylation are histone acetyl-transferases (HAT; for example, p300), which are lysine acetyl-transferases (KAT) that deposit acetyl groups in lysine residues of the histone tail. Conversely, lysine deacetylases (KDAC), including histone deacetylases (HDAC), can remove the acetyl groups from histones, thus compacting the chromatin. There are four classes of HDACs, Rpd3-like proteins (class I; HDAC1, HDAC2, HDAC3 and HDAC8), Hda1-like proteins (class II; HDAC4, HDAC5, HDAC6, HDAC7, HDAC9 and HDAC10), HDAC11 (class IV) and Sir2-like proteins (class III; SIRT1, SIRT2, SIRT3, SIRT4, SIRT5, SIRT6 and SIRT7).

Histone acetylation plays a dynamic role in the developmental-stage specific regulation of β-like globin gene expression (Figure 3). A positive correlation has been established between the β-like globin gene expression levels and H3 and H4 acetylation in the LCR and in the β-like globin genes in murine fetal liver cells and erythroid cell lines harboring the human β-globin locus [77,78,79].

Studies using human fetal and adult erythroid cells have shown that the levels of H3 acetylation in the LCR are stable during development, while the same marker in the β-globin coding regions positively correlates with *HBB* expression levels [65,80]. In general, in human primary cells, when a β-like globin gene is activated (e.g., the γ-globin gene in fetal erythroblasts), there is an H3 loss from the promoter and a strong acetylation of the remaining histones leading to a more open chromatin state [80]. In fact, human primitive-like, fetal and adult erythroid cells display H3 lysine 9 or 27 hyperacetylation in the ε-, γ- and β-globin gene promoters, respectively [55,66,67,80]

Interestingly, histone deacetylase activity is not always correlated with γ-globin repression. In fact, SIRT1, an histone deacetylase has been positively associated with increased γ-globin expression, and SIRT1 activators enhance the expression of γ-globin in primary human erythroid cells [81]. In fetal cord blood erythroid cells, the levels of SIRT1 are higher as compared to adult bone marrow cells [81]. SIRT1 was found to enhance the looping between the LCR and the γ-globin promoters, and this was also associated with an indirect increase in the levels of the activating H4K16ac mark in the γ-globin promoters [81]. Therefore, SIRT1 activators could potentially be used for reactivating HbF in hemoglobinopathy patients.

The interplay between different types of histone modifications also characterizes the β-globin locus regulation and has also been investigated at the level of enhancers (typically marked by H3K4me1) in human erythroid cell lines. In the β-globin LCR, H3K4me1 was correlated with histone acetylation, namely, H3K27ac. Interestingly, the loss of H3K4me1 led to a concomitant reduction in H3K27ac, but not vice versa, indicating a hierarchy that places methylation upstream of acetylation in the β-globin locus regulation [82].

Histone acetylations also cooperate with DNA methylation in regulating globin gene expression. In human cells, H3 and H4 acetylation regulates the β-globin genes in concert with other histone modifications (e.g., histone methylations and phosphorylation) and DNA methylation. Along the human erythroid differentiation, the activating H3K9ac marks in the β-globin locus overlap with histone methylation marks (H3K4me1 and H3K4me3), and 5-hmC [83], highlighting the interplay between different epigenetic marks to control gene expression. Furthermore, PRMT5 [64] or LYAR [74] downregulation in human erythroid cell lines led not only to the eradication of the H4R3me2s repressive mark, but also led to a reduction in the repressive H4 serine 1 phosphorylation (H4S1ph) mark and an increase in the activating H4K12ac, H4K8ac and H3K9ac marks. Inhibition of PRMT5 by Adox in adult bone marrow erythroid cells also increased the H4 acetylation levels at the γ-globin promoters [84].

### 2.3. The Interplay between Epigenetic Modifiers, Transcription Factors and Chromatin Looping

The fine-tuning of globin gene expression is mediated by the direct or indirect repressive or activating effects of TFs that bind to the β-globin locus, and by an interplay between these TFs and DNA or histone modifiers that alter the epigenetic profile of the locus and, thus, its chromatin accessibility. By way of example, the GATA1 TF interacts with the CBP/p300 acetyltransferase during erythropoiesis [85]. With regard to the fetal-to-adult Hb switching, the major players of this interplay are the BCL11A and LRF TFs that interact with co-factors including epigenome modifiers. BCL11A and LRF interact with the NuRD complex, which contains HDACs and MBD2. MBD2 binds to methylated CpG regions and recruits other NuRD components to repress gene expression [86]. BCL11A also interacts with histone demethylases, such as KDM1A, which removes activating histone methylation marks in the γ-globin promoters [43]. Another repressive complex that regulates the expression of γ-globin genes is the direct repeat erythroid-definitive (DRED) complex that contains KDM1A, DNMT1 and testicular receptor 2 and 4 (TR2/TR4)TF [43,87] (Figure 2).

Finally, the specific binding of TFs to the β-like globin genes and the chromatin remodeling through LCR-mediated looping formation is required for the regulated expression of β-like globin genes. The chromatin looping drives the γ-globin or β-globin gene expression in fetal and adult erythroblasts, respectively. The looping is mediated by a protein complex containing LDB1, LMO2, GATA1, TAL1 and E2A and facilitates the juxtaposition of the LCR to the γ- or the β-globin gene promoter, thus promoting their expression [32,33,34,35,36]. The recruitment of this protein complex inducing chromatin looping is also associated with epigenetic changes in the β-globin locus. In fact, in a murine erythroid cell line, GATA1 induces LDB1-mediated chromatin looping and H3K4me3 deposition at the β-globin gene to regulate its expression [88]. Furthermore, in a human fetal erythroid cell line, the ETO2 transcriptional co-repressor impairs the binding of this protein complex to the β-globin locus and, as a consequence, the looping between the LCR to the γ-globin gene. This is paralleled by the recruitment of the NuRD complex, the decrease in H3K27ac and H3K9ac at the β-globin locus, and the repression of the γ-globin genes [89]. Taken together, these results indicate a tight interplay between chromatin looping and epigenetic modifications at the β-globin locus to promote proper β-like globin gene regulation.

## 3. Epigenome Editing

Growing evidence of the crucial role of the epigenome in gene regulation, and the reversible nature of epigenetic modifications has led to the development of “epidrugs” as a clinical treatment for different diseases [90,91]. However, as they usually target epigenome modifiers, their effect is not sequence specific and can lead to a broad alteration of gene expression, sometimes leading to cell death [92].

In order to mitigate this effect, engineered epigenetic effectors that are capable of performing targeted epigenome editing have been developed. Targeted epigenome modifications can be generated by fusing an effector domain (that either has intrinsic epigenetic enzymatic activity or recruits epigenome modifiers in the case of TFs) to a sequence-specific DNA-binding domain (DBD) to ensure that the effector domain acts only on a specific target (Figure 4).

### 3.1. Zinc Finger-Based Epigenome Editors

Zinc-finger proteins (ZFs) have been the DBD of choice for the first epigenetic applications. Those proteins are composed of repetitive Cys2His2 domains, consisting of 28–30 amino acids, which allow binding to double-stranded DNA through a zinc ion, coordinated by four conserved residues [93]. Multiple fingers are joined together to form multi-ZF proteins, generating a versatile binding domain that binds DNA with very high affinity [94]. Indeed, each finger recognizes a specific 3 or 4 base pair sequence, allowing the binding to a specific target region. Those characteristics have been exploited to generate artificial polydactyl ZF proteins in vitro to target regions of interest [95].

Initially, ZFs were fused to TFs or epigenetic modifiers. They were coupled with TFs such as p65, viral activation domains such as VP16, or transcriptional activators composed of several tandem copies of VP16 to achieve activation of specific endogenous genes [96,97,98,99]. However, the use of these artificial TFs only leads to a modulation of the gene expression, since these tools might not efficiently modify the local epigenetic landscape, and their outcome can be influenced by the accessibility of the target site for the DBDs [100].

To overcome this limitation, a wide range of effector domains with intrinsic epigenetic activity have been employed in conjunction with ZF DBDs to either increase or repress gene expression [100,101,102,103]. Hence, ZFs have been fused to TET1 or TET2 domains to induce promoter demethylation, thus increasing gene expression [100,101], or to DNMT3A and its co-factor DNMT3L to de novo methylate promoters and decrease gene expression [102]. Moreover, the HAT domain of p300 has been fused to a ZF DBD to increase gene expression by depositing histone acetylation at promoter regions [103]. All these fusion proteins are characterized by a small size, which allows easy delivery by plasmid transfection [103] and retroviral vectors [104,105].

Although different platforms using ZFs as DBDs have been developed and have reached clinical application, the low specificity of the tools remains an important barrier for their use in the epigenome editing field. ChIP-seq analysis has showed that although ZFs preferentially bind to the target genomic region, there is also widespread binding at potentially off-target sites [99,101], most frequently mapping to promoter regions that show a significant sequence similarity with the intended target site [100]. Likewise, the addition of effector domains can alter the ZF binding distribution, increasing the off-target rate [106]. To overcome this issue, the effector domains have been divided into two separate components engineered with ZFs targeting adjacent regions. For instance, a monomeric methyltransferase was split into two fragments that cannot function unless assembled into an active heterodimeric enzyme. Each fragment was subsequently fused to a different zinc finger. This tool was used to drive CpG methylation at different target sites, resulting in a limited methyltransferase activity in the absence of the flanking zinc finger binding site [107]. However, it should be noted that protein complementation strategies remain challenging since they require complex protein engineering.

Lessons learned from ZF-based epigenome editors encouraged a faster development of other epigenome editing platforms with higher DNA-recognition capacities, namely, transcription activator-like effectors (TALE) and clustered regularly interspaced short palindromic repeats (CRISPR)/Cas9.

### 3.2. TALE-Based Epigenome Editors

An alternative DBD option for epigenome editing applications arose from the study of TALEs in bacterial plant pathogens [108]. These proteins present a modular DBD which can target any DNA region [109]. The TALE DBD is composed of 7 to 34 tandem repeats, each containing 33–35 amino acids [110]. These repeats are highly conserved, except at the hypervariable positions 12 and 13, the so-called repeat-variable diresidue (RVD) which determines the base preference of the repeat [111]). Thus, there is a 1:1 interaction between each TALE repeat and a single nucleotide. The specific binding properties of a RVD vary depending on the structure of the TALE. It is influenced by the total number of repeats, the position of the repeats within the TALE array and the surrounding RVD [112,113]. All these characteristics are deeply evaluated during the design of TALE DBDs for in vitro production to increase the specificity of the DBDs.

TALEs have been fused to various effector domains to either up- or down-regulate gene expression. TFs such as VP16 and VP64 were fused to TALEs with the aim of reactivating the pluripotency in somatic stem cells, by promoting the expression of SOX2, KLF4 and OCT4 [103,114,115,116].

Other artificial transcriptional repressors have been developed by combining TALEs with the catalytic domain of KRAB to silence endogenous gene expression. The KRAB domain has a transcriptional repressive activity and operates through interactions with different corepressors and chromatin modifiers that remove histone H3 acetylation and increase H3K9me3 [117]. TALE:KRAB fusions have been tested in HEK293FT cells to repress the *SOX2* gene [118], as well as in mice HSPCs to knock down the c-kit and *Pu.1* genes [119]. Moreover, TALEs fused with the KRAB domain have been described as an important tool to maintain pluripotency and self-renewal capacity in human HSPCs. In particular, TALE:KRAB has been successfully used to transiently silence the expression of *TCF3*, involved in the suppression of pluripotency and self-renewal pathways, thus providing a promising option for obtaining a sufficient number of HSCs for transplantation [120].

With the same intent of gene silencing, TALEs have also been tested in combination with other repressive domains. For instance, TALEs have been tethered with the mSin interaction domain (SID), which acts by reducing H3K9 acetylation, thus inducing transcriptional repression in cell lines (to repress the calcium channel related *CACNA1C* gene) [118] and primary murine neurons (to downregulate the expression of *Grm2* gene, encoding one of the major excitatory neurotransmitters) [121].

Several TALE-based tools have been developed to directly modulate DNA methylation or deposit or remove histone modifications. TALE-based epigenome editors have been used to remove aberrant DNA methylation, which is associated with several diseases. By way of example, TALE DBDs have been fused with TET1 to remove critical CpG methylations at the endogenous promoter of *KLF4* in the K562 cell line and reactivate its expression [101]. However, the role of CpG methylation in gene regulation is not always associated with gene repression especially when it occurs at enhancers or imprinting control regions [54]. Therefore, more recently, the TALE:TET1 epigenome editor has been used to target the imprinting control region 2 (ICR2) of p57 (a negative regulator of cell proliferation) to reduce its expression and stimulate the proliferation of β-cells, which can be used as replacement therapy in the context of type I and type II diabetes [122].

Other strategies with a focus on altering disease-associated epigenetic marks include TALE DBDs tethered with DNMTs to promote DNA methylation at specific loci. TALE:DNMT fusions have been used to silence the expression of *CDKN2A*, encoding the p16 protein that participates in cell senescence, with the aim of increasing the replication of primary human fibroblasts [123].

TALEs have also been fused with histone modifiers, such as KDM1A, which demethylates H3K4me2 and promotes the interaction with HDAC. When tested in a leukemia cell line, TALE:KDM1A was able to target an active H3K4me2+ and H3K27ac+ enhancer of the *SCL* gene (encoding for a transcription factor with critical function in hematopoiesis), and to significantly reduce histone methylation and acetylation, and as a consequence, *SCL* expression [124]. Furthermore, TALEs fused with HDAC8 were able to reduce acetylation of the *Grm2* promoter in primary neurons and consequently downregulate *Grm2*. However, when tested on other genes in different cell types, TALE:HDAC8 showed variable effects on gene repression, suggesting that the chromatin state influences the outcome [121].

Of note, several studies reported how the interaction between different epigenetic modifications play a crucial role in stable regulation of gene expression over time. In this regard, several studies combined different tools simultaneously introducing multiple epigenetic modifications. TALEs fused with KRAB, DNMT3A and DNMT3L have been individually tested to silence *B2M* expression in a leukemia cell line, obtaining no or transient effects. However, co-delivery of these three components significantly reduced *B2M* expression, and the effect was stable after numerous cell divisions. Moreover, the same platform was tested for silencing other endogenous genes, namely, interferon alpha/beta/omega receptor 1 (*IFNAR1*) and vascular endothelial growth factor A (*VEGFA*), confirming the crucial role of the interaction between epigenetic modifications in gene regulation [125].

Following these observations, other tools have been developed to perform combinatorial epigenetic modifications. Designer epigenome modifiers combining a single TALE with KRAB, DNMT3A and DNMT3L have been tested in primary human CD4^+^ T cells to efficiently silence *CCR5* and *CXCR4*. Interestingly, when splitting the KRAB and the DNMT activity into two distinct molecules, they failed to promote gene silencing [126].

Similarly to ZF DBDs, the off-target activity is one of the major concerns regarding TALE-based technology [101]. A number of studies have reported low but detectable off-target activity both in vitro and in vivo [113,127,128]. However, this depends on the specific TALEs, which can have differential off-target binding [113,129]. Moreover, a study showed that no changes in gene expression at off-target sites have been reported with TALE DBDs [130]. In addition, several improvements have been implemented to increase the binding specificity of TALE DBDs, such as the presence of divalent cations [131].

Another significant limitation of TALEs is their large number of tandem repeats, which makes TALE-expressing plasmid cloning and delivery difficult. However, after optimization and minimization of the number of tandem repeats, delivery as plasmid DNA or in vitro transcribed mRNA has been successful [126], as well as delivery with lentiviral [123], adenoviral [122] and adeno-associated viral (AAV) vectors. The latter are the preferred vectors for gene transfer in vivo, but have a limited packaging size, and thus, they can only express TALEs combined with functional domains of small size [121].

### 3.3. CRISPR/Cas9-Based Epigenome Editors

The CRISPR/Cas9 technology is commonly used for epigenome editing applications, thanks to its easier design and flexibility. The CRISPR/Cas9 system is based on a guide RNA (gRNA) which drives the Cas9 nuclease to the protospacer or target site, which has to be located upstream of a protospacer adjacent motif (PAM). The recognition of the target site is directed by the complementarity between the gRNA spacer and the DNA protospacer and allows the Cas9 nuclease to create a double-strand break (DSB) at the target site. Adaptations of Cas9 have been developed to create nuclease-deficient variants. For instance, the catalytically inactive dead Cas9 nuclease (dCas9 that does not generate DSBs but retains binding activity) has been fused to TFs or to epigenetic modifiers to modulate gene expression.

dCas9 was initially fused with VP64 and p65 (creating dCas9:VP64 and dCas9:p65) which were first used to target and activate GFP^+^ reporter gene in HEK293 cells [132]. The same tools have subsequently been used to target the promoter of endogenous genes, such as *VEGFA* and *NFT3,* increasing their expression in HEK293 cells [133]. KRAB domain has also been tethered to dCas9, and the first attempt was performed in HEK293 cells, whereby this tool efficiently reduced the expression of a GFP^+^ reporter gene [132]. More recently, dCas9:KRAB was used to develop a gene therapy strategy for retinitis pigmentosa. dCas9:KRAB was successfully delivered by AAV with the aim to repress the expression of *Nrl* gene, a master regulator of rod photoreceptor determination, which mediates the in situ reprogramming of rod cells into cone-like cells that are resistant to retinitis pigmentosa-specific mutations [134]. Moreover, dCas9:KRAB was shown to be able to cause potent and specific repression of *PTEN* (encoding a phosphatase that limits cell growth), in human cell lines and neural cells derived from human-induced pluripotent stem cells (iPSCs). Thus, suppression of *PTEN* resulted in a robust central nervous system (CNS) axon regrowth, providing a new option for promoting axon regeneration and functional recovery after CNS trauma [135].

Of note, experiments targeting both gene enhancers and promoters revealed that the design of the tools is critical for their activity. By way of example, the position of the fusion site between dCas9 and the TF, and the orientation of dCas9 and the effector domain can strongly impact the editing efficiency [132,136]. Moreover, dCas9:VP64 and dCas9:p300 have a lower efficiency compared to TALE:VP64 and TALE:p300, respectively, when targeting gene promoters, probably because the dCas9 fusions interfere with the binding of TFs [133,137].

The first studies aimed at optimizing the tools by increasing the number of transcriptional domains fused to the dCas9, leading to the generation of activators such as dCas9:VP160, which contains ten repeats of VP16 domain [138], or dCas9:VPR, where dCas9 was fused with a tripartite activator containing VP64, p65 and Rta [139]. More recently, dCas9 tethered with the VPR domain (MiniCAFE) was used to efficiently activate gene expression in *Caenorhabditis elegans*, and in murine and human cells [140].

However, a more stable epigenetic modification has been obtained by fusing dCas9 with domains of endogenous epigenome modifiers. dCas9 was initially fused to DNMT3A alone or in tandem with its cofactor DNMT3L, to obtain efficient site-specific DNA methylation and consequently transcriptional repression at several human genes. dCas9:DNMT3A, for instance, led to a therapeutic downregulation of *SNCA* in iPSCs from Parkinson’s disease patients, who typically present high *SNCA* levels [141]. Similarly, dCas9:DNMT3A-3L was used to target *CXCR4* and *TFRC* leading to gene repression in SKOV-3 and HEK293 cell lines [142]. To increase the stability of the epigenetic repression, a combination of this dCas9:DNMT3A and dCas9 fused with the KRAB domain was used to stably silence endogenous genes, namely, *VEGFA* and *IFNAR1* [125]. More recently, an optimized enzyme containing dCas9 fused with DNMT3A, DNMT3L and KRAB domain was successfully used to repress gene expression in iPSCs and other cell types and maintain a stable repression in iPSC-derived neurons [143].

To reactivate gene expression, CRISPR/Cas9-based epigenome editors capable of demethylating DNA have been developed. To this end, TET1 has been tethered to dCas9 targeting the promoter of *BDNF* in post-mitotic neurons, a gene that encodes a protein involved in neuronal survival and differentiation, leading to reduced DNA methylation and *BDNF* re-expression [144]. The same tool has been used in the context of fragile X syndrome (FXS), a disease characterized by a silencing of the *FMR1* gene associated with hypermethylation of the CGG expanded repeats in the 5′UTR of *FMR1*. dCas9:TET1 led to efficient demethylation of the CGG repeats and to a consequent *FMR1* reactivation in vivo in mice [145]. dCas9:TET1 has also been used to demethylate the promoter of the onco-suppressor *BRCA1,* which is often hypermethylated in breast cancer cells. In this case, dCas9:TET1 increased *BRCA1* expression and inhibited the proliferation of a cancer cell line, highlighting the therapeutic potential of epigenome editing to reduce hypermethylation in promoters of tumor suppressor genes, which is one of the hallmarks of cancer [146]. dCas9:TET1 has also been used in Jurkat T cells to drive the demethylation of a regulatory region (FOXP3-TSDR; Treg specific demethylated region) of FOXP3, a master transcription factor of regulatory T (Treg) cells. Consequently, this approach might represent a tool for programming of primary T cells toward the Treg phenotype [147]. However, another study in primary murine T-lymphocytes proved that demethylation of the *Foxp3* enhancer alone was insufficient to stabilize the *Foxp3* expression and modulation of histone modifications was also necessary to activate the gene [148]. Finally, dCas9:TET1 has been used to study the role of TET1 in cardiac progenitor differentiation from murine embryonic stem cells (ESCs) [149]. Targeting the promoter of *Hand1* (encoding a factor that plays a key role in cardiac development) using dCas9:TET1 up-regulated *Hand1* expression, it was shown that TET1 and DNA methylation play a crucial role in regulating gene expression during cardiac differentiation [149].

CRISPR/Cas9-based epigenome editors able to modulate histone modifications have been generated by fusing dCas9 with different histone modifiers. For instance, the catalytic core domain of the human HAT p300 protein was fused to dCas9 to deposit H3K27 acetylation at both promoters and enhancers (*IL1RN, MYOD* and *OCT4* gene promoters and HS2 enhancer of the β-globin locus) and activate gene expression in HEK293T cells [103,130,140]. More recently, dCas9:p300 has been used in the context of cystic fibrosis leading to elevated expression of the *CFTR* gene (affected in this disease) with a consequent restoration of the normal phenotype in patient-derived human bronchial epithelial cells when combined with a pharmaceutical treatment [150].

Conversely, targeted removal of histone acetylation can be achieved using HDAC fused with dCas9 resulting in gene repression, although one study has highlighted the importance of cellular context-dependent effects when using HDAC-containing epigenome editors. This work showed that histone deacetylation by some CRISPR/Cas9 epigenome editors can drive either gene repression or activation depending on the chromatin context and cell type. In particular, dCas9:HDAC3 was used to remove H3K27ac from the *Mecp2* promoter in two murine cell lines expressing different *Mecp2* levels; this led to activation or repression of gene expression according to the cell-specific chromatin state of the promoter [151].

Deactivation of enhancers and gene repression can likewise be achieved by using dCas9 fused to the histone demethylase KDM1A, which eliminates H3K4me2, a histone mark typically found at active promoters and enhancers. A first attempt to use dCas9:KDM1A was performed in murine ESCs, targeting the enhancers of genes that encode factors critical for maintaining the stem cell state (*Oct4* and *Sox2)*; this led to the loss of expression of these genes and consequent phenotypic changes [152]. Recently, dCas9:KDM1A has been used to target cis-regulatory elements in chicken embryos and alter the gene expression [153]. In this study, a comparison of dCas9:KDM1A and dCas9:KRAB suggests a more efficient silencing by KDM1A when used at early stages of gene activation, whereas KRAB-mediated repression might be more potent with genes already engaged in transcription [153].

H3K9me3 and H3K27me3 histone modifications can also be deposited at target loci using dCas9 fused with Suppressor of variegation 3–9 homolog 1(SUV39H1) and EZH2, respectively. However, these studies showed that a repressive histone modification alone is not sufficient for gene silencing [154]. Indeed, recent approaches combined different epigenetic modifications to achieve a more predictable and persistent silencing; e.g., strategies combining Cas9:EZH2d and DNMT3 to achieve gene silencing [155].

Overall, CRISPR/Cas9-based epigenome editors present several advantages compared to ZFs and TALEs. The main advantage of the CRISPR/Cas9 system is the ease of designing a single guide RNA (sgRNA) targeting a specific region compared to the protein design required for TALE- and ZF-based epigenome editors. Off-target binding can also occur with the CRISPR/Cas9 system [156]. However, when off-target binding was observed, changes in gene expression or even in epigenetic marks and chromatin accessibility were not detected outside of the intended target site [103,136,155]. Furthermore, CRISPR/Cas9 off-target effects can be reduced to undetectable levels with high-fidelity Cas9 variants. In particular, different versions of Cas9 have been developed, in which residues that normally make nonspecific contacts with the DNA are mutated [157]. These Cas9 variants have low or no off-target activity, while maintaining a high on-target efficiency [157,158]. Finally, the use of the CRISPR/Cas9 system facilitates simultaneous epigenome editing of multiple regions with the delivery of only one Cas9 enzyme [138,159].

CRISPR/Cas9-based epigenome editors have been successfully delivered into cells using plasmid [125], RNA [126], ribonucleoprotein [160] and lentiviral vectors [161]. However, the use of Cas9 DBDs would require vectors of large capacity for in vivo applications. Because of their limited packaging size, AAVs cannot accommodate dCas9 fused with the effector domain. To overcome this limitation, Cas9-based epigenome editors have been split into two subdomains and delivered using two different AAVs [134]. Alternatively, recently engineered smaller Cas orthologs can be accommodated in AAVs. Finding an appropriate delivery method remains crucial in developing Cas9-based epigenome editing in vivo.

### 3.4. Epigenetic Approaches to Modulate β-like Globin Expression

The importance of epigenetic regulations on globin gene expression has fueled the identification of novel medications capable of directly altering the epigenome for therapeutic purposes. Epigenetic regulation can act through multiple mechanisms that eventually lead to chromatin alterations that either permit or inhibit transcription [162]. Both DNA methylation and histone modifications, such as acetylation and methylation, play a crucial role in regulating the expression of β-globin chains [55,163].

For example, DNA methylation of the CpG dinucleotides at the *HBG* promoters negatively correlates with γ-globin gene expression in primary erythroblasts [55,57]. Furthermore, *HBG* promoters are hypomethylated in individuals harboring HPFH mutations [58]. As a consequence, DNA methyltransferase (DNMT) inhibitors, such as decitabine, have been used to treat patients with β-thalassemia and SCD. Clinical trials showed that decitabine is well tolerated and is efficacious in increasing HbF levels in both β-thalassemia [164] and SCD [165] patients. However, decitabine also had the effect of increasing platelet counts and has therefore yet to be approved as a treatment for the diseases due to concerns regarding its adverse effects. New compounds, such as GSK3482364, are currently being tested in pre-clinical studies [166].

Activating histone modifications, such as H3K27ac or H3K4me3, mark active *HBG* promoters in fetal but not in adult erythroblasts. HDAC inhibitors, such as vorinostat, have been used in clinical trials with the aim of reactivating HbF [167]. Vorinostat is a pan-HDAC inhibitor approved for the treatment of T-cell lymphoma and significantly upregulates γ-globin expression in erythroid cell lines and primary human HSPC-derived erythroblasts. However, the clinical trial conducted to evaluate its efficacy in SCF patients was terminated early due to poor patient recruitment. By contrast, a specific class I HDAC inhibitor leads to HbF reactivation and increased acetylation of H3 and H4 in the γ-globin promoters in erythroid cells derived from treated SCD HSPCs [168], and it could be further tested in clinical studies.

In addition, other drugs interfering with the activity of epigenetic modifiers modulating histone methylations have been used to reactivate HbF and will likely be further developed, such as RN-1, an inhibitor of KDM1A [169,170,171], UNC0638, an inhibitor of EHMT1/2 [70,71], and FTX-6058, an inhibitor of EED [172].

However, the mode of action of these pharmacological inhibitors are often uncertain as they are non-specific, and their global effect poses serious safety concerns. For instance, down-regulation of components of chromatin-remodeling and modifier complexes (e.g., HDACs, DNMT1) via shRNA or CRISPR/Cas9 is associated with HbF upregulation [43,173]. However, in many cases, this led to impaired cell fitness [174] as these factors play a pleiotropic role on the establishment and maintenance of the epigenetic and transcriptional cellular profiles. For the same reason, the mechanisms underlying HbF reactivation upon down-regulation of these factors are still not clear.

Several gene therapy strategies for reactivating the expression of the endogenous fetal γ-globin have been successfully developed. However, although they offer the opportunity to achieve permanent therapeutic changes in the genome, these approaches still rely on the intermediate formation of DSBs. This in turn poses serious safety concerns as the introduction of DSBs might result in unpredictable genotoxic effects [175]. Thus, a technology capable of silencing the expression of a target gene without altering the underlying genomic sequence is desirable.

Artificial transcription factors and epigenome editors have been used to directly reactivate human γ-globin or β-globin expression and to study the mechanisms of β-like globin gene regulation. The use of an artificial ZF transcriptional activator ZF:VP64 targeting the *HBG* promoters successfully increased HbF expression in primary human erythroid cells [98] and an in vivo study also demonstrated increased γ-globin gene expression in the peripheral blood from β-YAC transgenic mice stably expressing the artificial ZF transcriptional activator [176]. Nonetheless, the use of this tool was associated with some cell toxicity [98]. Fusion of the TET1 demethylase to a TALE DBD targeted to the β-globin promoter efficiently reactivated the *HBB* gene through targeted DNA demethylation in an HEK293 cell line [133].

Both dCas9:p300 [102] and dCas9:KRAB were used to target the HS2 enhancer, which regulates the expression of β-like globin genes. While dCas9:p300 caused H3K27 acetylation and activation of globin gene expression in HEK293T cells [103], dCas9:KRAB-induced H3K9 trimethylation at the HS2 enhancer and a consequent reduction in chromatin accessibility at the enhancer and globin gene promoters, demonstrating that repression mediated by dCas9:KRAB is able to reduce enhancer activity via local modification of the epigenome [177].

However, since epigenetic regulation at the globin loci is complex, it is reasonable to believe that the fine-tuned expression of the different genes requires the interplay of different epigenetic modifications in parallel to the recruitment of TFs. The first attempt to both deposit histone modifications and recruit TFs was performed by taking advantage of the natural interaction that occurs between the phage MS2 hairpins and MS2 bacteriophage coat protein (MCP). In particular, dCas9:p300 was fused to MS2-sgRNA sequence which recruits the VP64 activator domains fused with MCP and targets the HS2 enhancer. This allowed the concomitant introduction of H3K27ac, the recruitment of a transcriptional activator and, as a consequence, the potent activation of β-like globin genes in HEK293T cells [178].

## 4. Conclusions and Perspectives

The regulation of β-like globin gene expression has been extensively studied, which has led the identification of different layers of regulation, involving transcription factors, chromatin looping and epigenome modifications of the β-globin locus. The interplay between TFs, DNA methylation and histone modifications play a crucial role in determining gene activation or repression.

The availability of genome editing tools has allowed the development of therapeutic approaches for β-hemoglobinopathies through the direct genetic modification of HSCs that either correct the disease-causing mutation or reactivate fetal hemoglobin expression by altering transcription factor occupancy. However, genome editing tools can cause a DNA damage response and unwanted genomic alterations. By contrast, epigenome editing would allow the modulation of gene expression (e.g., re-activation of *HBG* genes) in a scarless manner, i.e., without affecting genome integrity. Moreover, epigenome editing is more targeted as compared to drugs able to change the epigenetic profile of the β-globin locus (such as HDAC and DNMT inhibitors).

Insights for the usage of epigenome editing as a therapeutic approach for hemoglobinopathies have been provided in cell lines and primary erythroid cells, through the therapeutic reactivation of HbF. Nevertheless, there are several steps that need to be taken before such an approach moves to a clinical phase. For instance, further studies are required in order to achieve persistent gene activation after transient exposure to the epigenome modifier, and the persistence of epigenome modifications (preferably without the constitutive expression of the epigenetic modulators) needs to be evaluated in long-term repopulating of patient HSCs. Furthermore, an extensive assessment of the off-target activity of epigenome modifiers is crucial, and last but not least, it is vital that clinically relevant protocols are developed for the delivery of this system.

## Figures and Tables

**Figure 1 genes-14-00577-f001:**
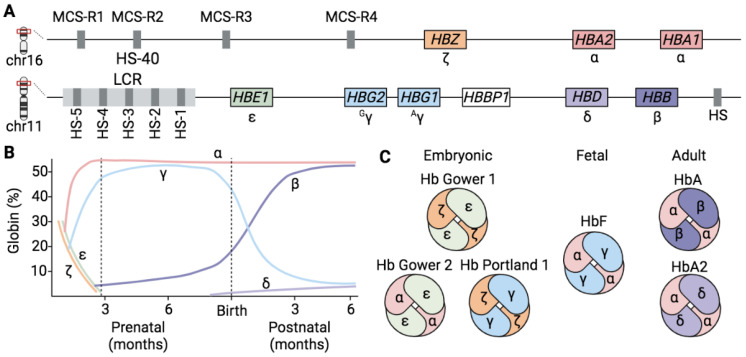
Human α- and β-globin loci and hemoglobin switching. (**A**) Schematic representation of the α- and β-globin loci on chromosomes 16 and 11, respectively. α-like (*HBZ*, *HBA2* and *HBA1*) and β-like (*HBE1*, *HBG2*, *HBG1*, *HBD* and *HBB*) globin genes are depicted with colored boxes in 5′ to 3′ orientation. The globin encoded by each gene is reported below each box. *HBBP1* pseudogene is depicted with a white box. The HSs are presented with grey boxes. (**B**) Diagrammatic representation of the human hemoglobin embryonic-to-fetal and fetal-to-adult switching, describing the expression pattern of α-like (α and ζ) and β-like (ε, γ, β and δ) globin chains, from conception to birth, and to adult life. (**C**) Composition of hemoglobin for different developmental stages (embryonic, fetal and adult Hb). Chromosome (Chr); multispecies conserves sequences (MCS); locus control region (LCR); hypersensitive site (HS); hemoglobin (Hb); fetal hemoglobin (HbF); adult hemoglobin (HbA); adult hemoglobin 2 (HbA2). Created with BioRender.com.

**Figure 2 genes-14-00577-f002:**
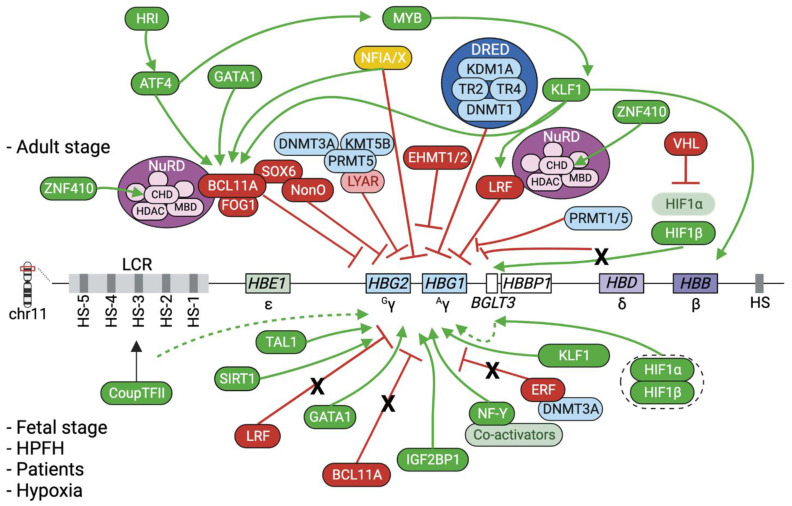
Regulation of β-like globin gene expression. Schematic model of β-like globin gene regulation in the adult (upper panel) and fetal stages, in individuals with hereditary persistence of fetal hemoglobin (HPFH) mutations, or in hemoglobinopathy patients, or in low O_2_ conditions (lower panel). Numerous transcription factors have been identified to regulate γ-globin expression in fetal/adult stages or HPFH conditions, either by directly silencing (NFIA/X [15], ERF [16]) or activating (Coup-TFII [17], IGF2BP1 [18], HIF1α/β [19], NF-Y [20,21], GATA1 [20] and KLF1 [22]) γ-globin genes, or by co-acting with or indirectly regulating the expression of the two major γ-globin repressors BCL11A (KLF1 [23], NFIA/X [15], ATF4 [24,25], SOX6 [26]) and LRF (KLF1 [27]) or other key-players such as KLF1 (MYB [28]), ATF4 (HRI [24,25]), MYB (ATF4 [29]) and CHD4 (ZNF410 [30,31]). Ovals represent proteins with repressive (red), activating (green), repressive/activating (yellow) activity, or epigenetic modifying activity (blue and purple). Dotted green lines show indirect activation and dotted black ovals indicate protein heterodimers. Created with BioRender.com.

**Figure 3 genes-14-00577-f003:**
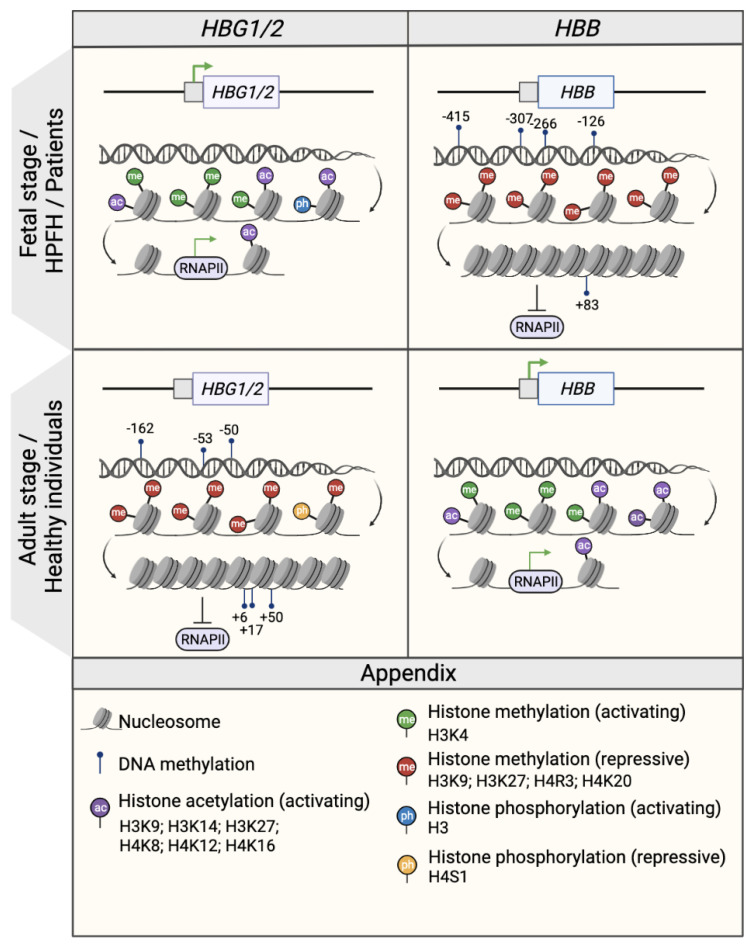
Epigenetic regulation of the β- and γ- globin gene expression. Schematic representation of the epigenetic profile of *HBG1/2* and *HBB* promoters (represented by squares) in the fetal or fetal-like stages (fetal life, individuals with HPFH mutations and β-hemoglobinopathy patients; upper panel) and in the adult stage (healthy individuals; middle panel). The presence of DNA methylation and histone modifications (activating acetylation marks and activating or repressive methylation marks) is associated with an open or closed chromatin conformation that influences the accessibility of RNA polymerase II (RNAPII) and, consequently, the transcription rate as a whole. Green arrows indicate active transcription. CpG sites that are specifically methylated in the *HBG1/2* genes in the adult stage are indicated and named after their distance from the transcription start site (e.g., −162, −53, −50, +6, +17, and +50). Created with BioRender.com.

**Figure 4 genes-14-00577-f004:**
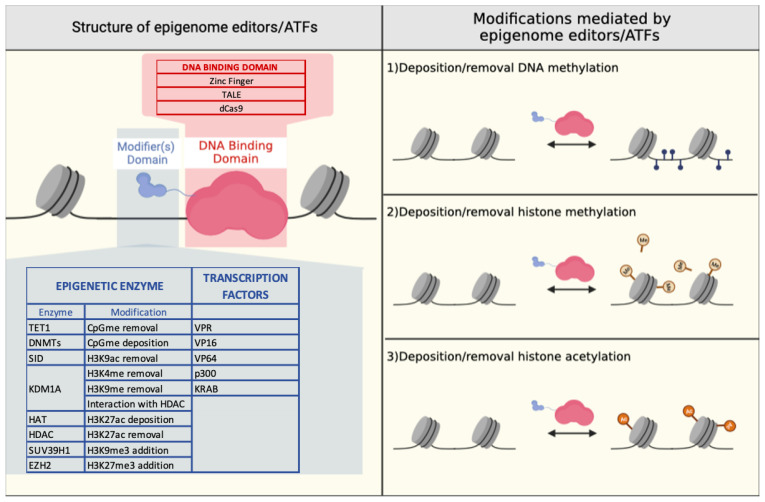
Overview of epigenome editors and artificial transcription factors: domains and epigenetic modifications. Schematic representation of epigenome editors and artificial transcription factors: (left panel). DNA-binding domains (DBDs) (zinc-finger, transcription activator-like effector (TALE) or dead Cas9 (dCas9)) can be fused with TFs or one or more epigenome modifiers. The epigenome modifier is an enzymatic domain that removes or adds epigenetic marks. The transcription factor recruits other activating/repressing co-factors depositing or removing epigenetic marks. The domains and related epigenetic modifications are listed in the table. Schematic representation of epigenetic modifications mediated directly or indirectly by epigenetic modifiers and TFs (right panel). Blue dots represent DNA methylation of CpG dinucleotides, and yellow dots represent histone methylation and orange dots represent histone acetylation. CpG methylation (CpGme); trimethylation of lysine 9 of histone 3 (H3K9me3); acetylation of lysine 9 of histone 3 (H3K9ac); methylation of lysine 4 of histone 3 (H3K4me); methylation of lysine 9 of histone 3 (H3K9me); acetylation of lysine 27 of histone 3 (H3K27ac); Histone deacetylase (HDAC); Ten-eleven translocation methylcytosine dioxygenase 1 (TET1); DNA (cytosine-5)-methyltransferase (DNMT); Krüppel associated box (KRAB); mSin3 interaction domain (SID); Lysine-specific histone demethylase 1A (KDM1A); histone acetyltransferase (HAT); Suppressor of variegation 3-9 homolog 1 (SUV39H1); Enhancer of zeste homolog 2 (EZH2). Created with BioRender.com.

## Data Availability

Not applicable.

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
