# Peer review of "Epigenetic Regulation of β-Globin Genes and the Potential to Treat Hemoglobinopathies through Epigenome Editing"

_genes, 2023, doi:10.3390/genes14030577_

Round 1
Reviewer 1 Report
The review by Fontana et al. discusses the epigenetic mechanisms of beta-like globin gene regulation and their manipulation as potential therapeutic approaches in treating hemoglobinopathies. The main concept of the review is sound and timely and would definitely be of interest to researchers in the field. However, there are several serious issues with the review in its present format. First, the review is too long, including sections that are not of direct relevance to the main topic, thus losing its focus and impact. Also, some aspects of the review are dealt with rather superficially and are lacking synthesis, for example, the fetal-to-adult Hb switching. The level of detail varies widely and is not always justified. For example, the background description of epigenetic modifications has been previously covered exhaustively by many others and should be a lot more comprehensive and focused for the purposes of this review. By contrast, there is practically no background given for important platform technologies in epigenome editing, such as TALENs and CRISPR-Cas9. There is also some repetition in the review, for example epigenetic inhibitors and gamma-globin reactivation are discussed under the epigenetic modifications’ sections and under Epigenetic approaches to modulate beta-like globin expression. The manuscript also needs to be edited for English. A small but indicative example is the extensive use of the “On the contrary” instead of the correct “By contrast”.
Other points:
1. Figure 2: figure resolution needs to be improved. Also, dotted lines need to be explained in the legend and some of the names of the proteins in the figure are not clearly visible due to the use of white font against a light-colored background (e.g., Co-activators, HIF1a etc.).
2. Figure 3: figure resolution needs to be improved. Overall, this figure is rather generic and does not add much.
3. Convention of writing mouse or human gene names should be consistently adhered to throughout the text.
4. It is not clear to me what is meant by “Natural” epigenetic modulation used in the title and in the main text.
Author Response
Reviewer 1
Comments and Suggestions for Authors
The review by Fontana et al. discusses the epigenetic mechanisms of beta-like globin gene regulation and their manipulation as potential therapeutic approaches in treating hemoglobinopathies. The main concept of the review is sound and timely and would definitely be of interest to researchers in the field. However, there are several serious issues with the review in its present format. First, the review is too long, including sections that are not of direct relevance to the main topic, thus losing its focus and impact. Also, some aspects of the review are dealt with rather superficially and are lacking synthesis, for example, the fetal-to-adult Hb switching. The level of detail varies widely and is not always justified. For example, the background description of epigenetic modifications has been previously covered exhaustively by many others and should be a lot more comprehensive and focused for the purposes of this review. By contrast, there is practically no background given for important platform technologies in epigenome editing, such as TALENs and CRISPR-Cas9. There is also some repetition in the review, for example epigenetic inhibitors and gamma-globin reactivation are discussed under the epigenetic modifications’ sections and under Epigenetic approaches to modulate beta-like globin expression. The manuscript also needs to be edited for English. A small but indicative example is the extensive use of the “On the contrary” instead of the correct “By contrast”.
We have reduced the length of the review by eliminating sections that are not of direct relevance. We also provided background information on ZF, TALE and CRISPR nucleases. We have checked for repetition throughout the manuscript and removed all the repetitive information. English was revised.
Other points:
- Figure 2: figure resolution needs to be improved. Also, dotted lines need to be explained in the legend and some of the names of the proteins in the figure are not clearly visible due to the use of white font against a light-colored background (e.g., Co-activators, HIF1a etc.).
We have now provided the figure with a better resolution, explained dotted lines in the legend and modified the white font.
- Figure 3: figure resolution needs to be improved. Overall, this figure is rather generic and does not add much.
We have now provided a high-resolution figure 3.
- Convention of writing mouse or human gene names should be consistently adhered to throughout the text.
We have now modified the mouse or human gene names accordingly throughout the text.
- It is not clear to me what is meant by “Natural” epigenetic modulation used in the title and in the main text.
We have now changed the title and the text.
Reviewer 2 Report
The review by Fontana and colleagues tackles the topic of the β-globin regulation. It is a well written review, with extensive details and maybe somewhat too long. The authors tackle various topics related to the β-globin regulation. Maybe the authors could add a section about how chromatin looping interplays with epigenetic modulation to promote the proper the β-globin regulation.
Author Response
Reviewer 2
Comments and Suggestions for Authors
The review by Fontana and colleagues tackles the topic of the β-globin regulation. It is a well written review, with extensive details and maybe somewhat too long. The authors tackle various topics related to the β-globin regulation. Maybe the authors could add a section about how chromatin looping interplays with epigenetic modulation to promote the proper the β-globin regulation.
We have reduced the length of the review by eliminating sections that are not of direct relevance. We have also provided additional information regarding the interplay of chromatin looping and epigenetic modulation in the paragraph “The interplay between epigenetic modifiers, transcription factors and chromatin looping”.
Reviewer 3 Report
Beta-globin gene disorders are the main complications of hematological disorders. The authors described the "epigenetic modulation of β-globin gene 2 expressions" which are new strategies for treatment and management in this group of disorders. The manuscript is well described and details the problem.
Author Response
Reviewer 3
Comments and Suggestions for Authors
Beta-globin gene disorders are the main complications of hematological disorders. The authors described the "epigenetic modulation of β-globin gene 2 expressions" which are new strategies for treatment and management in this group of disorders. The manuscript is well described and details the problem.
We thank the reviewer for appreciating our manuscript.